# Microstructure Development in Artificially Cemented, Fine-Grained Soils

Simon Oberhollenzer [1,*], Andre Baldermann [2], Roman Marte [1], Djemil Mahamat Moussa Tahir [1], Franz Tschuchnigg [1], Martin Dietzel [2] and Manfred Nachtnebel [3]

1 Institute of Soil Mechanics, Foundation Engineering and Computational Geotechnics, Graz University of Technology, Rechbauerstraße 12, 8010 Graz, Austria
2 Institute of Applied Geosciences, Graz University of Technology, NAWI Graz Geocenter, Rechbauerstraße 12, 8010 Graz, Austria
3 FELMI-ZFE, Steyrergasse 17, 8010 Graz, Austria
* Correspondence: s.oberhollenzer@tugraz.at

**Abstract:** Fine-grained sedimentary deposits can bear an increased risk for building settlements due to their moderate stiffness and strength properties, as well as high groundwater tables. However, some buildings, e.g., situated on shallow foundations in Alpine basins, show only relatively small settlements because the formation of carbonate cement can create bridging bonds between the detrital soil particles, leading to increased stiffness. These weak bonds can be damaged through dynamic loads and high static loads, causing a weakening of the soil's microstructure and resulting in large settlements in several cases. However, the environmental controls and mechanistic processes underlying the formation versus damaging of microstructure in fine-grained, postglacial sediments are, to date, poorly understood. In the present study, fine-grained sediments are artificially cemented by calcium carbonates ($CaCO_3$) to investigate (i) the influence of a mild and sustainable cementation process on the stress–strain behavior of silicate- and carbonate-rich soils and (ii) the possibilities and limitations of artificial microstructure development for soil stabilization. Incremental load oedometer testing (*IL*), bender element testing (*BE*), X-ray diffraction (*XRD*), scanning electron microscopy (*SEM*) and Brunauer–Emmett–Teller (BET) specific surface area (SSA) measurements are used to characterize the development of cementation and to elucidate the improvements in the soil mechanical properties. It is shown that cementation induced by $CaCO_3$ mineralization (by 5–15% replacement) leads to an increased stiffness (factor $\approx$ 5–7) and shear wave velocity (factor $\approx$ 1.1), caused by the formation of nanocrystalline, particle-binding $CaCO_3$ cements. The improvement of soil stiffness is dependent on the $CaCO_3$ replacement level, reaction time and primary soil mineralogical composition.

**Keywords:** artificial cementation; soil microfabric; microstructure development; stiffness; soil stabilization; calcium carbonate

## 1. Introduction

Basins and valleys of the Alpine region, which formed during several glacial-interglacial periods of the Pleistocene, were filled by lake and riverine sediments over thousands of years. These geologically young sediments are usually characterized by a high groundwater table and appear in a normally consolidated state, thus providing the base for today's urban areas, infrastructure and agricultural uses. However, especially fine-grained, calcareous and siliciclastic deposits found in various areas in Europe and worldwide can lead to an increased risk for building settlements due to their poorly developed stiffness and strength properties [1,2]. Those unfavorable soil mechanical properties often come along with high extra costs for foundation and construction works or lead to frequent damages caused by building settlements [3]. For instance, it has been estimated that damages in clayey soils increased from USD 2.2 billion (1973) to USD 15 billion (2012) across the United States over the last decades [4,5].

Chemical stabilization techniques using cement, lime, fly ash, calcium binders, polymers and granulated blast-furnace slag have been used with different success in these soils [6–14]. As an alternative, mechanical stabilization techniques, such as the installation of water barriers and geo-membranes, can be applied [15–18]. However, chemical and mechanical stabilization techniques can be harmful to the environment and often result in high costs [19] since the production of 1 t cement or lime causes up to 1.2 t of carbon dioxide ($CO_2$) emissions [20–22]. Recently, microbial-induced calcium carbonate precipitation (*MICP*) has been represented as an environmental-friendly, bio-mediated soil-improvement technology [23,24]. For example, the hydrolysis of urea enables the bio-inspired precipitation of calcium carbonate ($CaCO_3$) minerals as a sustainable alternative for soil stabilization, which has ~90% less abiotic depletion potential and 3% less global warming potential compared to traditional Portland cement applications in soil stabilization [25]. It was shown by several authors that *MICP* can mitigate seismic-induced soil liquefaction and reduce the soil's permeability, compressibility, and shear strength [26–36]. Consequently, bio-inspired geotechnical engineering is becoming increasingly important nowadays [37].

In contrast, the authigenic formation of $CaCO_3$ phases can be observed in many natural surroundings using mineralogical, hydrochemical and microstructural testing [38–45]. These naturally formed cement phases can fill the pore space, increase the bonding between detrital and other authigenic particles and thus lead to an increased soil stiffness and strength due to the development of a microstructure. Consequently, multi-story buildings founded on shallow foundations often show relatively small settlements even under high static loading. Otherwise, dynamic loads induced by heavy construction measures, such as soil improvement measures via jet grouting, can lead to increased settlements, as grain-to-grain contacts and interparticle bonding is partly or fully destroyed [3]. In this context, the term *fabric* is used to describe the arrangement of particles, particle groups and pore spaces in soils, while the term *structure* describes the entire particle arrangements (fabric) and their stabilities [46,47]. Thus, *(micro)structure* is often used synonymously with the soil's bonding strength or interparticle forces in the literature [48]. Two soils characterized by the same fabric can have different microstructural properties (mainly related to stiffness and strength) if interparticle forces differ. Until now, systematic experimental studies on fine-grained, postglacial sediments that aim at elucidating the generic links between the formation of a microstructure and the development of relevant soil mechanical parameters (i.e., stiffness) are scarce. Further, the combined influences of reaction time, cementation type and degree and soil mineralogy are often not considered when microstructure development is considered on-site.

In the present paper, fine-grained soil specimens are artificially cemented by nanocrystalline $CaCO_3$ binders using sodium hydrogen carbonate ($NaHCO_3$) and calcium oxide ($CaO$) as reactants in order to investigate the potential of sustainable carbonate cementation for stabilizing soft, fine-grained sediments with calcareous versus siliciclastic composition. Therefore, oedometer and bender element tests were executed on cemented specimens, and the results were compared with non-cemented, untreated (reference) materials. Mineralogical and microstructural investigations by X-ray diffraction (*XRD*), scanning electron microscopy (*SEM*) and Brunauer–Emmett–Teller (*BET*) surface area analyses are further used to visualize differences in mineralogy and fabric upon $CaCO_3$ mineralization and cementation. The possibilities and application limits of soil improvement using this mild and sustainable cementation procedure are discussed.

## 2. Materials

The mineralogical composition of the materials used here resembles those found in the different Alpine basins from Austria and adjacent regions, which are characterized either by mixtures of siliciclastics and carbonates (e.g., clay minerals, feldspar, quartz and dolomite) or almost pure carbonates (predominantly calcite). Here and in the following, we use the term 'soil' analogous to fine-grained, postglacial sediments. For example, the fine-grained sediments from the Rhine valley and Seekirchen basin are dominated by calcite, while an

increased amount of clay minerals and other silicate phases is recognizable in the basin of Salzburg (Austria). Accordingly, two defined mineral assemblages (*M1* and *M2*) were chosen for laboratory testing, which reflectf the two main soil compositions mentioned before: *M1* is mainly composed of calcite (93%) with minor muscovite (5%) and quartz (2%); *M2* contains a mixture of dolomite (50%), muscovite and chlorite (10%), quartz (33%) and albite (7%), as determined by prior mineralogical analysis of the standard materials used for the preparation of the mixtures *M1* and *M2* (as quality control).

The raw calcite (0–200 µm) and dolomite (0–100 µm) used for laboratory testing were provided by the *DOLOMIT Eberstein Neuper GmbH*. To account for muscovite, quartz and feldspar within *M2*, the dolomite raw material was mixed with the embankment material of a trial embankment from an Austrian basin in the ratio of 50/50. Since both testing materials (*M1*, *M2*) should present a particle size distribution (*PSD*) similar to that of postglacial (Alpine) sediments, calcite and dolomite were crushed in several rounds using a rock crusher. As indicated by the blue and red dashed lines in Figure 1, the particle size distribution (*PSD*) of both materials changed significantly during the crushing procedure. The *PSD* of the testing material, presented in Figure 1 by solid lines, is classified according to ISO 14688-1 [49] as sandy clay-silt mixture (*M1*: *sa' Cl/Si*) and sandy clayey silts (*M2*: *sa' cl Si*), respectively (see below for methodology). In addition to their respective *PSDs*, results of particle density and Atterberg limits (*ATT*) are summarized for both materials in Table 1.

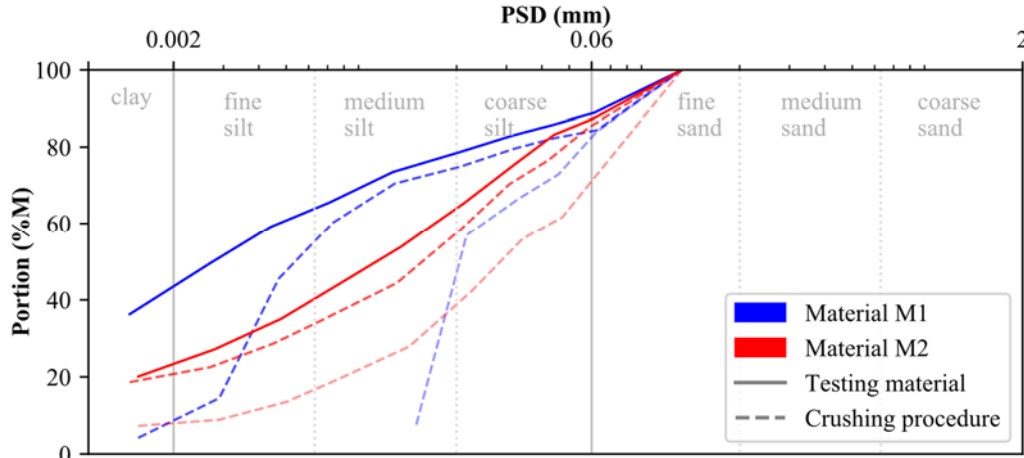

**Figure 1.** Particle size distribution of materials *M1* and *M2*.

**Table 1.** Characterization of raw materials *M1* and *M2* with respect to their soil classification, particle size distribution, particle density and Atterberg limits.

| | Material *M1* | Material *M2* |
|---|---|---|
| ISO 14688-1 | *sa' Cl/Si* | *sa' cl Si* |
| USCS (ASTM D2487-11) | *CL* | *CL* |
| Clay content (<0.002 mm) | 43.5 | 22.1 |
| Silt content (0.002–0.063 mm) | 45.8 | 65.9 |
| Sand content (0.063–2 mm) | 10.7 | 12.0 |
| Particle density $\rho_s$ (g/cm$^3$) | 2.69 | 2.76 |
| Liquid limit *LL* (%) | 35.9 | 36.0 |
| Plastic limit *PL* (%) | 20.7 | 20.3 |
| Plasticity index *PI* (%) | 15.2 | 15.7 |

While material *M2* is characterized by a higher particle density ($\rho_s$) than material *M1*, the liquid limit (*LL*), the plastic limit (*PL*) and the plasticity index (*PI*) are in good agreement between both materials. The Casagrande results of both mineralogies are situated above the A-line within the range of medium plasticity clays, leading to a lean clay (*CL*) classification according to USCS (ASTM D2487-11 [50]). Reference is made to Section 4 for further information on the methods used.

### 3. Artificial Cementation and Sample Preparation

The artificial cementation (i.e., the accelerated formation of $CaCO_3$ binder phases to enhance microstructure development) of materials *M1* and *M2* was induced by introducing a mixture of calcium oxide ($CaO$) and sodium hydrogen carbonate ($NaHCO_3$) powders at different weight percent ratios to the raw materials *M1* and *M2*. Afterward, adequate amounts of distilled water were added to the mixtures, which led to the dissolution of the easily soluble salts and subsequently resulted in the fast formation of $CaCO_3$ under the development of moderately alkaline conditions (see the simplified Equation (1)). Compared to authigenic $CaCO_3$ mineralization and cementation frequently observed in Alpine post-glacial sediments, the carbonation reaction used here is faster and more efficient. Moreover, this procedure is more sustainable and environmentally friendly compared to common soil stabilization measures utilized, e.g., cement injections, where portlandite ($Ca(OH)_2$) is formed under extremely alkaline conditions, causing negative impacts to soil biota and the aquatic environment.

$$CaO + NaHCO_3 \rightarrow CaCO_3 + NaOH \tag{1}$$

To study the influence of newly formed $CaCO_3$ cements on the stress–strain behavior of fine-grained soils, soil specimens were prepared at different degrees of cementation. Even though soil specimens differ in terms of cementation, uniform densities and water contents were considered during specimen preparation ($\rho_d = 1.40$ g/cm$^3$, $\rho_{sat} = 1.92$ g/cm$^3$, $w = 35.0\%$, $S_r = 100\%$), resembling common Alpine soils. The present approach considers a replacement of 5%, 10% and 15% of the *M1* and *M2* raw materials (with respect to their dry masses) by $CaO$ and $NaHCO_3$ additives, which were added in stoichiometric ratios following Equation (1). Reaction times of 1 and 3 weeks were considered for each replacement step and both soil mixtures.

In detail, the dried material (*M1* or *M2*) was mixed with $CaO$ and $NaHCO_3$ salts in the first step at the aforementioned solid–solid ratios. After homogenization in a mixer for 1 min in a dry state, adequate amounts of distilled water were added to each mix design, and the mixing procedure was continued until a defined water content was reached along the homogenized soil specimens. Subsequently, the soil–water mixtures were carefully transferred into plexiglass cylinders ($h = 46$ mm, $d = 100$ mm)—situated on top of a wrapped wood plate—using a lab spatula (Figure 2). A constant loading pressure was used to reach a homogeneous density along the specimens and to minimize the entrapment of air inclusions. After specimen preparation (Figure 2), the density was verified by weighing with recognition of the volume of the plexiglass cylinder. Subsequently, a second wooden plate was added to the top of the samples and pressed against the cylinder to minimize water evaporation and sample drying out during the experiment (Figure 2d).

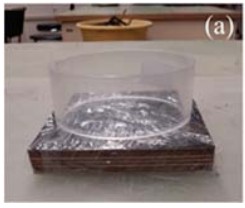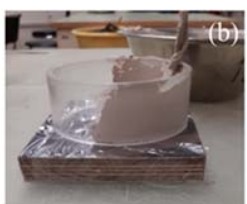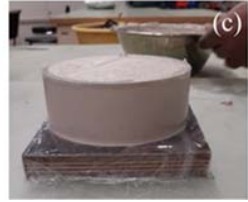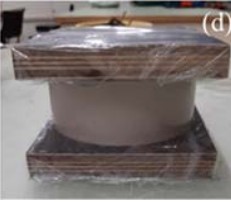

**Figure 2.** Sample preparation for oedometer testing of uncemented and artificially cemented soil specimens: Summary of individual working steps (**a**–**d**).

All soil specimens were stored at a constant temperature of $22 \pm 1$ °C throughout the reaction time of 1 to 3 weeks. Afterward, the plexiglass cylinders were removed, and oedometer rings were gently pushed into the soil specimens. The remaining material was used for *XRD*, *SEM* and *BET* tests. Larger cylindrical shells ($d = 100$ mm, $h = 220$ mm) were used for triaxial and bender element testing. An additional latex membrane (placed between the soil sample and cylinder) was used to avoid any sticking of the material along the shells. After the reaction time, the soil specimens were slightly shortened to $h = 190$ mm and used for triaxial and bender element testing.

## 4. Methods

### 4.1. Soil Mechanical Testing

Oedometer tests enable the characterization of the stress–strain behavior of soil samples under axial loading whilst preventing radial deformations. Therefore, a soil specimen of cylindrical shape (diameter $d$ = 7 cm and height $h$ = 2 cm) was placed within a steel ring and loaded vertically. Porous filter discs were placed at the bottom and at the top of the soil specimen to allow pore water dissipation during loading. The applied vertical load and the deformations were measured manually or automatically during test execution. To reduce the influence of wall friction (soil–ring interaction) on measured settlements, the diameter to height ratio $d/h$ was set to >2.5 for all investigations [51]. During the incremental loading (*IL*) testing procedure, the load application was performed fully automated by electric or pneumatic presses. This stress-controlled procedure is based on changing the vertical load incrementally. The following load steps were chosen in the present study: 10, 20, 40, 80, 160, 320, 640, 1000, 320 and 20 kPa. For each load-step, the vertical stress was held constant until consolidation is completed (normally after 24 h). All *IL* oedometer tests were performed according to ISO 17892–5 [51].

Bender elements (*BE*) consist of two piezoceramic bimorph sheets (e.g., zirconate titanate and lead titanate) with external conducting surfaces sandwiching a conductive center shim (e.g., ferrous nickel) [52]. Thereby, the lack of crystal symmetry or the electrically polar nature of the crystals results in piezoelectricity [53]. These electromechanical transducers enable the conversion of mechanical to electrical energy and vice versa. If a driving voltage is applied to the bender elements, they bend due to the polarization and act as a transmitter. On the other hand, if these elements are mechanically forced to bend, voltage is generated (receiver). Here, parallel type bender elements with a dimension of 12 × 12 × 1.2 (length × width × thickness in millimeters) and an additional polyurethane coating were used and placed within a triaxial device. The configuration shown in Figure 3 was chosen for the present study, where a function generator produces a defined input voltage that is sent to the oscilloscope and the transmitter bender element (situated within the top cap of the triaxial device).

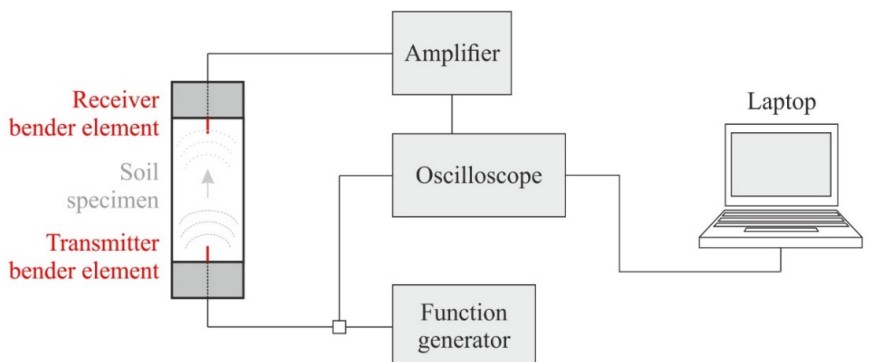

**Figure 3.** Bender element test set-up within a triaxial device (modified according to [54]).

The chosen frequency of the excitation voltage is strongly influenced by the soil stiffness and the particle size distribution, usually ranging from 1 to 30 kHz in soils. While the arrival time is not affected by the signal type and frequency, the ability to detect the signal can change strongly [55]. Based on this test configuration, the shear wave velocity $V_S$ can be calculated according to Equation (2) based on the tip-to-tip travel length $L_{tt}$ between the two bender elements and the (measured) shear wave travel time $t$.

$$V_S = \frac{L_{tt}}{t} \tag{2}$$

Even though bender elements are increasingly used in geotechnical engineering, no standardized procedure is accepted for defining the arrival time $t$ [52,53,56]. As recom-

mended by various authors [53,56], the first reversal and the first reversal after the first zero were chosen for the detection of the shear wave arrival time.

After installing the soil specimen in the triaxial device, the specimens were isotropically consolidated to $p'_1$ = 50 kPa, $p'_2$ = 150 kPa and $p'_3$ = 300 kPa. The shear wave velocity ($V_S$) was determined after each (isotropic) consolidation stage. No saturation phase was initially defined since the saturation degree ($S_r$) was larger than 96% for all soil specimens. After isotropic consolidation, an undrained shear phase was started, and $V_S$ was measured in intervals between 3 and 5 h until failure was reached. A sine wave (amplitude = 12 V) with frequencies ranging from 1 to 30 kHz was used to detect the arrival time.

Soil classification was executed according to ISO 14688–1 [49] and ASTM D2487-11 [50] (Unified Soil Classification System = USCS). Therefore, the soil particle size distribution (*PSD*) and the Atterberg limits (*ATT*) were determined according to ISO 17892-4 [57] and ISO 17892–12 [58], respectively. Physical indices of soil specimens, such as the water content (*w*), wet density ($p_{unsat}$), dry density ($p_d$) and particle density ($p_s$), were determined based on ISO 17892-1 [59], ISO 17892-2 [60] and ISO 17892-3 [61], respectively.

*4.2. Mineralogical and Microstructural Testing*

The mineralogical composition of the powdered, dried soil samples was determined through quantitative X-ray diffraction (*XRD*) analyses [62]. In the present work, a PANalytical X'Pert PRO diffractometer (Co-Kα radiation source) operated at 40 kV and 40 mA and outfitted with a high-speed Scientific X'Celerator detector was applied for the characterization of soil specimens. The soil powders were prepared by the top-loading technique and examined in the range 4–85° 2θ with a step size of 0.008° 2θ and a scan speed of 40 s per step [2]. For mineral identification and quantification, the PANalytical X'Pert Highscore Plus software and its implemented ICSD database were used, with an analytical error of <3 wt% per mineral phase [63].

The visualization of the soil (micro)fabric before and after the carbonate cementation was realized via a scanning electron microscopy (*SEM*) study. Therefore, representative soil samples were cut perpendicular to the bedding plane and carbon-coated to reduce charging. Backscattered electron images, secondary electron images and element-specific X-ray radiation spectra were collected, which provide information about the shape, size and distribution of individual particle assemblages, pore system and the nature of the distinct mineral phases present in the soil specimens. In the present work, FEI Quanta 450 FEG (FELMI-ZE, Graz) and Zeiss DSM 982 Gemini (University of Graz) microscopes were used, respectively. Different magnifications were used to visualize the soil fabric at macro- and micro-scales [64,65].

The Brunauer–Emmett–Teller (*BET*) method was used to determine the specific surface area (*SSA*) [66] of the soil specimens (pre-dried at 40 °C), utilizing nitrogen ($N_2$) gas at relative gas pressure ratios ($p/p_0$) from 0.05 to 0.35 (Micrometrics FlowSorb II 2300). A multi-point adsorption method was used to quantify the impact and progress of $CaCO_3$ cementation on the pore size distribution of the soil samples, with an estimated analytical error of ±10% [67]. The *SSA* of the soil specimens was determined according to Equation (3):

$$\frac{1}{X \cdot [(p_0/p) - 1]} = \frac{1}{X_m \cdot C^*} + \frac{C^* - 1}{X_m \cdot C^*} \cdot \left( \frac{p}{p_0} \right) \tag{3}$$

where $X_m$ is the number of gas molecules needed to form a monolayer, $X$ represents the adsorbed gas molecules at a given relative pressure ($p/p_0$) and $C^*$ is a constant related to the heat of adsorption. Based on a linear regression, $X_m$ can be computed and used to determine the specific surface area $a_{s,BET}$ (m²/g). The pore volume and the pore size distribution of the specimens were determined by incrementally increasing the gas pressure until all pores are filled. The evaluation of the adsorption and desorption isotherms was made using the *BJH* (Barrett, Joyner and Halenda) calculations [68]. The flowchart, shown in Figure 4, summarizes the individual steps of laboratory testing.

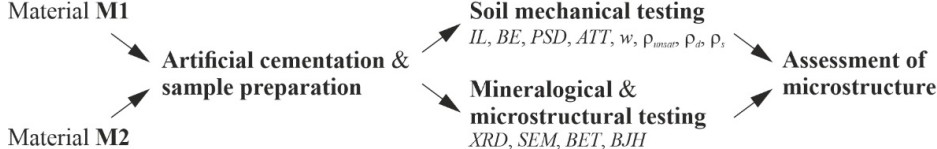

**Figure 4.** Summary of individual work steps for laboratory testing (flowchart).

## 5. Results and Discussion

### 5.1. Influence of Artificial Cementation on 1 D-Compression

The stress–strain behavior of the uncemented and artificially cemented soil specimens was investigated by means of oedometer tests, and the results are displayed in Figure 5 for both materials (*M1* and *M2*). It is evident that the mineralogical composition (carbonate- versus silicate-dominated samples), the degree of cementation (5% to 15% replacement by $CaO + NaHCO_3$) and the reaction time applied (1 to 3 weeks) have an anticipated effect on the stress–strain behavior of artificially cemented soil specimens. In detail, the uncemented samples (black, dotted lines in Figure 5) present an almost linear trend within the log $\sigma'_v$–$\varepsilon_{vol}$ space for primary loading stress paths, which indicates the lack of microstructure in the untreated soil specimens [46,47]. Notably, increased apparent preconsolidation pressures [69] develop for both materials (*M1*, *M2*) as a function of the degree of cementation (blue (5%), red (10%) and green (15%) lines in Figure 5) and with the increase in time (cf. dashed and solid lines in Figure 5), which is due to a higher degree of $CaCO_3$ precipitation. The latter binds and interconnect individual detrital soil particles, thus forming calcite- cemented aggregate structures, which lead to increased stiffness. These observations are in good agreement with previous oedometer test results obtained on both artificially cemented soils [70,71] and natural (structured) clays [47,72–74].

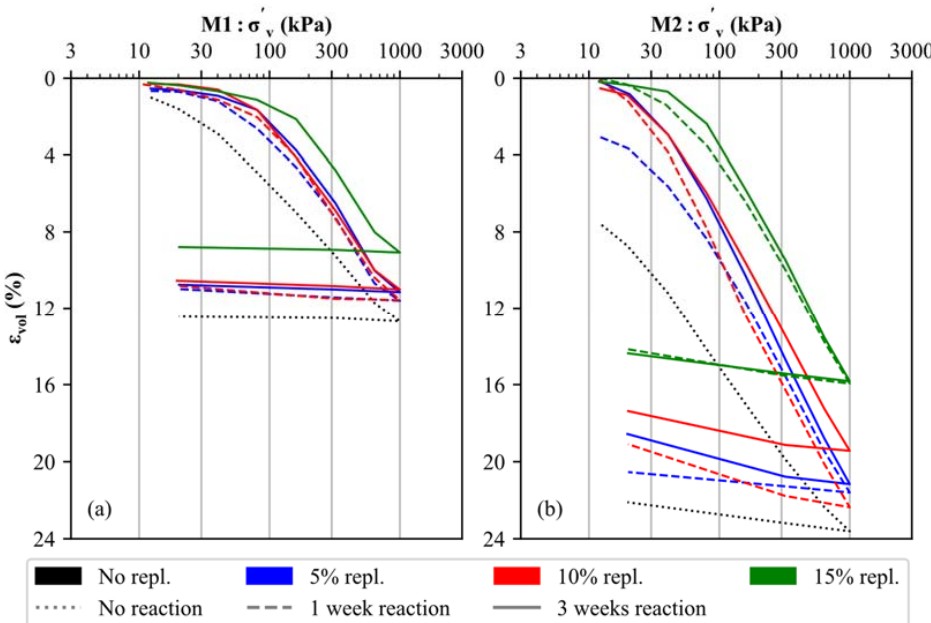

**Figure 5.** Results of oedometer tests executed on uncemented and artificially cemented soil specimens: (**a**) *M1* and (**b**) *M2*.

As long as interparticle bonds are intact (i.e., the apparent preconsolidation pressure is not reached), a stiffer behavior is observed for all cemented samples compared to the uncemented samples. After exceeding this stress level, the stiffness of all soil samples decreases, indicating a breakdown of the calcite-cemented aggregate structures in both materials [75]. While cementation at 5% and 10% substitution leads to a slightly improved but generally similar stiffness for *M1* and *M2*, respectively, a significantly higher apparent preconsolidation pressure is observed for 15% material replacement (green lines in Figure 5),

which indicates the formation of a stronger microstructure. Similarly, the apparent preconsolidation pressures for the 5% and 10% replacement samples amount to approximately 60–90 kPa (*M1*) and 20–40 kPa (*M2*), respectively. Increased pressures of approximately 180 kPa (*M1*) and 60 kPa (*M2*) can be derived for the 15% replacement samples. For all degrees of cementation, an increased reaction time (3 weeks—solid line) leads to a slightly stiffer behavior.

By comparing the oedometer test results of materials *M1* and *M2*, it can be observed that the calcite-dominated soil specimens are, in all cases, characterized by a stiffer behavior, even though *M1* is characterized by a finer *PSD* (Figure 1). It can therefore be assumed that the primary mineralogical composition of any soil material has a significant influence on its stress–strain behavior (for materials with and without artificial cementation).

The dry and wet densities, as well as the water contents, which were determined before load application, are reported for materials *M1* and *M2*, with recognition of variable degrees of carbonate cementation, in Table 2. It is evident that the dry and wet densities of both materials slightly increase with the rising formation of $CaCO_3$ phases, which can be attributed to a carbonate (micro) filler effect that comes along with an increase in the packing density of the soil particle assemblages (as discussed in detail below).

**Table 2.** Summary of measured wet densities ($p_{unsat}$), dry densities ($p_d$) and water contents ($w$) of soil specimens from the *M1* and *M2* series used for oedometer testing.

| Soil Specimen | Wet Density $p_{unsat}$ (g/cm$^3$) | Dry Density $p_d$ (g/cm$^3$) | Water Content $w$ (%) |
|---|---|---|---|
| M1/No repl. | 1.89 | 1.40 | 35.00 |
| M1/5% repl./1 week | 1.89 | 1.44 | 31.65 |
| M1/5% repl./3 weeks | 1.92 | 1.45 | 32.42 |
| M1/10% repl./1 week | 1.90 | 1.44 | 31.95 |
| M1/10% repl./3 weeks | 1.89 | 1.42 | 32.52 |
| M1/15% repl./1 week | 1.92 | 1.45 | 32.63 |
| M1/15% repl./3 weeks | 1.92 | 1.45 | 32.13 |
| M2/No repl. | 1.92 | 1.42 | 35.00 |
| M2/5% repl./1 week | 1.90 | 1.41 | 34.58 |
| M2/5% repl./3 weeks | 1.92 | 1.43 | 34.51 |
| M2/10% repl./1 week | 1.93 | 1.44 | 34.16 |
| M2/10% repl./3 weeks | 1.91 | 1.41 | 34.75 |
| M2/15% repl./1 week | 1.93 | 1.45 | 33.64 |
| M2/15% repl./3 weeks | 1.95 | 1.46 | 33.57 |

Similar observations have been reported for $CaCO_3$-supported hydrated cement blends, which showed higher strength values at increased carbonate (micro/meso) filler contents [76,77]. Simultaneously, the water content generally shows a reverse trend for all soil samples, i.e., a decrease can be seen at higher cementation levels, even though a saturation degree ($S_r$) > 97 was ensured in all specimens. The reason for this observation is discussed in Section 5.3. Due to differences in the density between the uncemented and cemented soil specimens, oedometer curves do not coincide at large strains in Figure 5.

### 5.2. Influence of Artificial Cementation on the Shear Wave Velocity

The results of the triaxial and bender element tests for the uncemented and cemented specimens are displayed in Figure 6. As an example, the curves are shown and compared for materials *M1* and *M2* without cementation and with 10% replacement, respectively, considering a reaction time of 3 weeks. It is evident that the cemented specimens (red curves in Figure 6a,b) show slightly higher deviatoric stresses ($q$) during undrained shearing compared to the uncemented specimens (cf. black curves in Figure 6a,b). However, we note here that the cemented specimen of *M1* was not sheared to failure to avoid bender element damage. Furthermore, material *M1* is characterized by a higher shear strength than *M2*. As discussed in more detail below, no peak shear strength is observed for the cemented

specimens as the main destructuration occurred during isotropic consolidation. The latter statement is confirmed by oedometer results shown in Figure 5, where all specimens display an apparent preconsolidation pressure smaller than 200 kPa. As positive excess pore water pressures develop during undrained shearing, contractive behavior is observed for both materials. Consequently, the mean effective stress ($p'$) decreases during shearing while the deviatoric stress ($q$) rises. As shown in Figure 6c,d, the cemented and uncemented soil specimens basically follow the same trend, as destructuration mainly occurs before the shear phase.

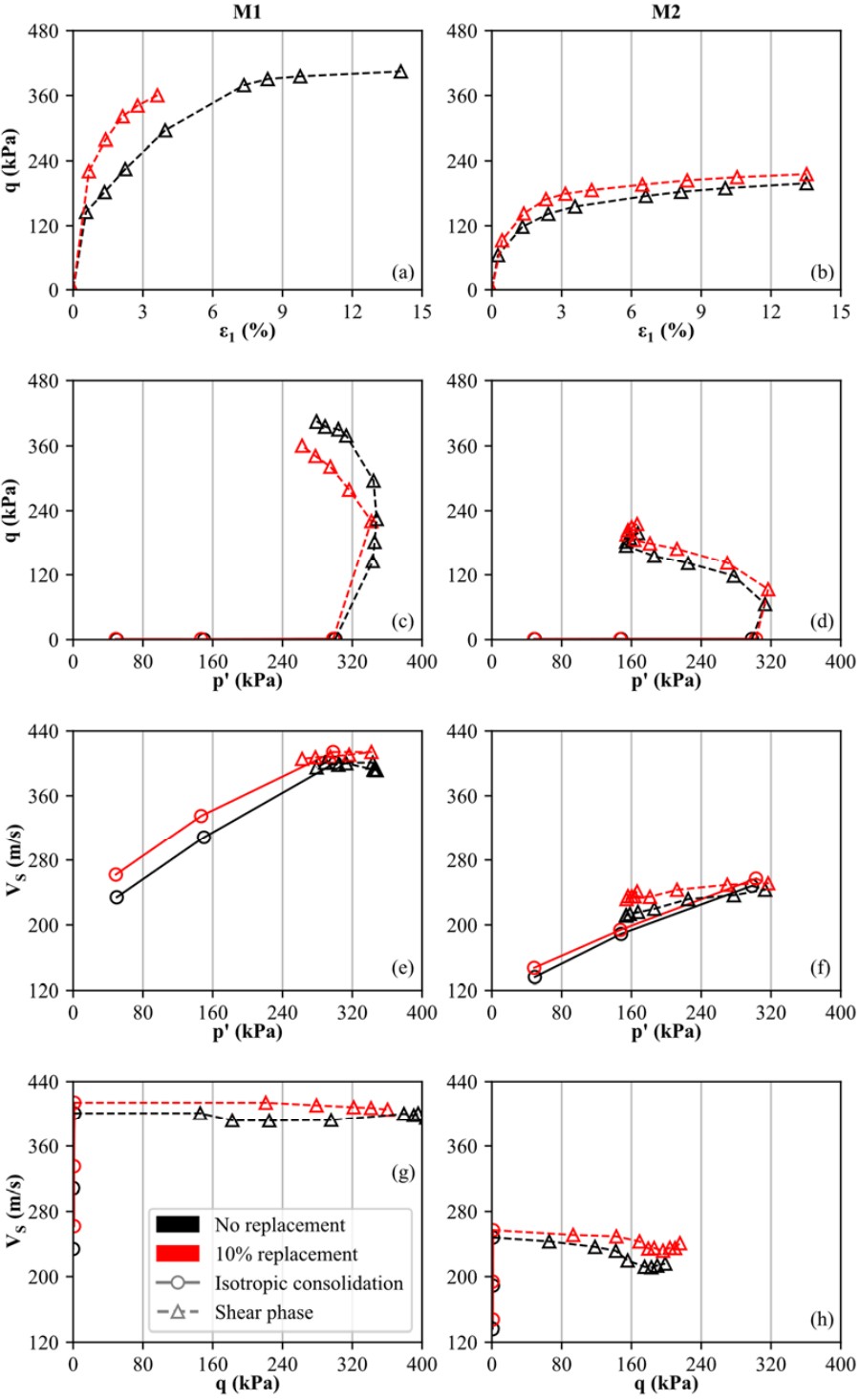

**Figure 6.** Results of triaxial and bender element tests, executed on uncemented and artificially cemented soil specimens: (*M1:* (**a**,**c**,**e**,**g**) column, *M2:* (**b**,**d**,**f**,**h**) column).

To study the influence of cementation (at 10% replacement) on the specimen's shear wave velocity ($V_S$), the mean effective stress $p'$ and the deviatoric stress $q$ are compared to $V_S$ in Figure 6e–h. The isotropic consolidation stages are indicated by solid lines and circles, whereas the results from the shear phase are represented by dashed lines and triangles. It is observed (as expected) that $V_S$ rises with increasing mean effective stress ($p'$) during isotropic consolidation (black and red lines in Figure 6e,f). This trend is in good agreement with in-situ measurements of seismic flat dilatometer tests (*SDMT*) in normally-consolidated soils, where $V_S$ increases over depth [73,74,78].

At defined stress levels ($p'_1 = 50$ kPa, $p'_2 = 150$ kPa, $p'_3 = 300$ kPa), the cemented specimens display higher $V_S$ values compared to the uncemented samples, even though the difference between the cemented and uncemented specimens decreases slightly with destructuration (increasing $p'$). Since the absolute $V_S$ measurements are higher for *M1* compared to *M2,* it is confirmed that the carbonate-dominated soil is characterized by a stiffer behavior. Furthermore, a slightly stronger fabric is assumed for *M1* since the difference between the cemented and uncemented specimens is slightly larger compared to *M2*. However, for $p'_1 = 50$ kPa, the ratio of cemented to uncemented shear wave velocity does not exceed 1.2 for both materials. In contrast, bender element results of artificially cemented sands (using the *MICP* technique) lead to ratios of >2.5 for similar calcite contents [79]. From Figure 6g,h, it can be seen that the change in shear wave velocity is small within the shear phase as $p'$ and $q$ show the reversed trend. As the decrease in $p'$ is smaller compared to the increase in $q$, it is confirmed that $p'$ has a stronger influence on $V_S$ than $q$.

### 5.3. $CaCO_3$ Precipitation and Microstructure Development in Artificial Soil Samples

The nature of the precipitated cement phases was studied for materials *M1* and *M2* by means of *XRD* analyses to understand the above-described differences in the mechanical behavior of uncemented and artificially cemented soil specimens. The mineralogy of material *M1* is mainly composed of calcite and, to a smaller extent, clay minerals (muscovite with traces of chlorite) and quartz (Figure 7a), while material *M2* contains mainly quartz, clay minerals (muscovite and minor chlorite), feldspar (albite and orthoclase), dolomite and traces of calcite (Figure 7b), as stated in the *Materials* section. Upon artificial cementation, the diffracted intensity of the calcite peaks increased notably in Figure 7, which indicates that this phase is the main reaction product in both materials and thus contributes to the observed microstructure development.

The other mineral phases (originally present in *M1* and *M2*) were not significantly affected by this chemical treatment method. While the amount of clay minerals remains constant for different degrees of cementation, the quartz content is slightly decreasing due to dilution effects induced by $CaCO_3$ formation. With the increasing degree of cementation, minor amounts of the hydrated sodium-calcium carbonate mineral pirssonite ($Na_2 Ca(CO_3)_2 \cdot 2 H_2O$) and trace amounts of portlandite ($Ca(OH)_2$) formed, in addition to massive calcite precipitation, at 5% and 15% replacement (Figure 7). The formation of such hydrated phases could explain the observed decrease in the soil specimen's water content at higher replacement levels, as their precipitation consumes water (Table 2). However, if these phases contribute to the development of a microstructure in both soil samples, uncertainty remains, and further research is required. The results of scanning electron microscopy (*SEM*) confirms that newly formed calcite cements have a stabilizing effect on the fabric of both materials (Figure 8). The fabric of the uncemented specimens is presented in Figure 8a,b for materials *M1* and *M2*, respectively. By comparing the *SEM* images, it becomes obvious that *M1* is characterized by a finer *PSD* than *M2*. Consequently, the *SEM* results are in good agreement with the *PSD* results shown in Figure 1.

A uniform distribution of large, single pores is indicated by the blue ellipses for both materials. As for the cemented specimens (Figure 8c,h), selected areas filled by authigenic carbonates (i.e., calcite) are marked by red dashed lines. Since the raw material of *M2* consists only of traces of calcite, the abundant $CaCO_3$ phases are all newly formed during

the cementation phase, covering large parts of the original sample. For *M1*, mixtures of newly formed and existing calcite are indicated by the red dashed lines. However, the cement phases are nano- to microcrystalline in all cases and present no rhomboid crystal shapes, which can be attributed to comparably short reaction times of 1 or 3 weeks, i.e., documenting the fast and massive formation of *CaCO₃* phases. We note here that we cannot fully exclude that $CaCO_3$ formation progressed through the instantaneous precipitation of highly reactive amorphous calcium carbonates (*ACC*), which subsequently transformed into the more stable anhydrous mineral form, calcite [65]. In contrast, earlier studies showed that calcium carbonates, formed by means of *MICP*, are larger in size and present a rhomboid shape [80]. Importantly, the amount of particle-interconnecting *CaCO₃* cements notably increases at higher replacement levels, giving rise to the formation of a microstructure via detrital particle clogging, pore space reduction and carbonate filler effects [77]. Consequently, the embedding of existing primary particles increases uniformly, which subsequently leads to an increase in wet and dry density, as summarized in Table 2.

The above-described carbonate (nano/micro) filler effect of generating a microstructure in materials *M1* and *M2* upon cementation is further evidenced by our $N_2$-*BET* physisorption study. Figure 9 highlights the physico-mechanical modifications of the specific surface area $a_{s,BET}$ and pore size distribution exemplarily for the uncemented and cemented soil specimens of material *M1*. Since the newly formed, nanocrystalline $CaCO_3$ cements reduce the open pore space, the $a_{s,BET}$ values decrease with increasing degree of cementation (Figure 9a). It is further confirmed that $a_{s,BET}$ is more strongly influenced by the replacement level compared to the reaction time (1 week or 3 weeks).

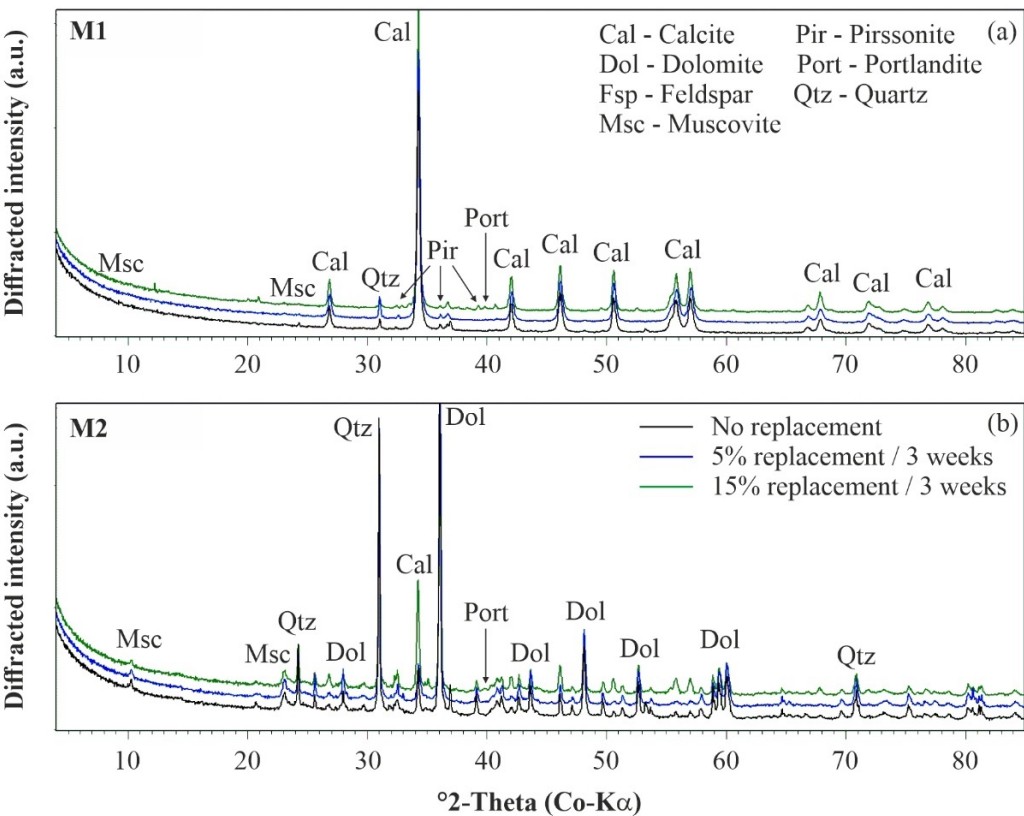

**Figure 7.** Mineralogy of uncemented and artificially cemented soil specimens: (**a**) *M1* and (**b**) *M2*. *XRD* patterns have been shifted vertically for better visibility.

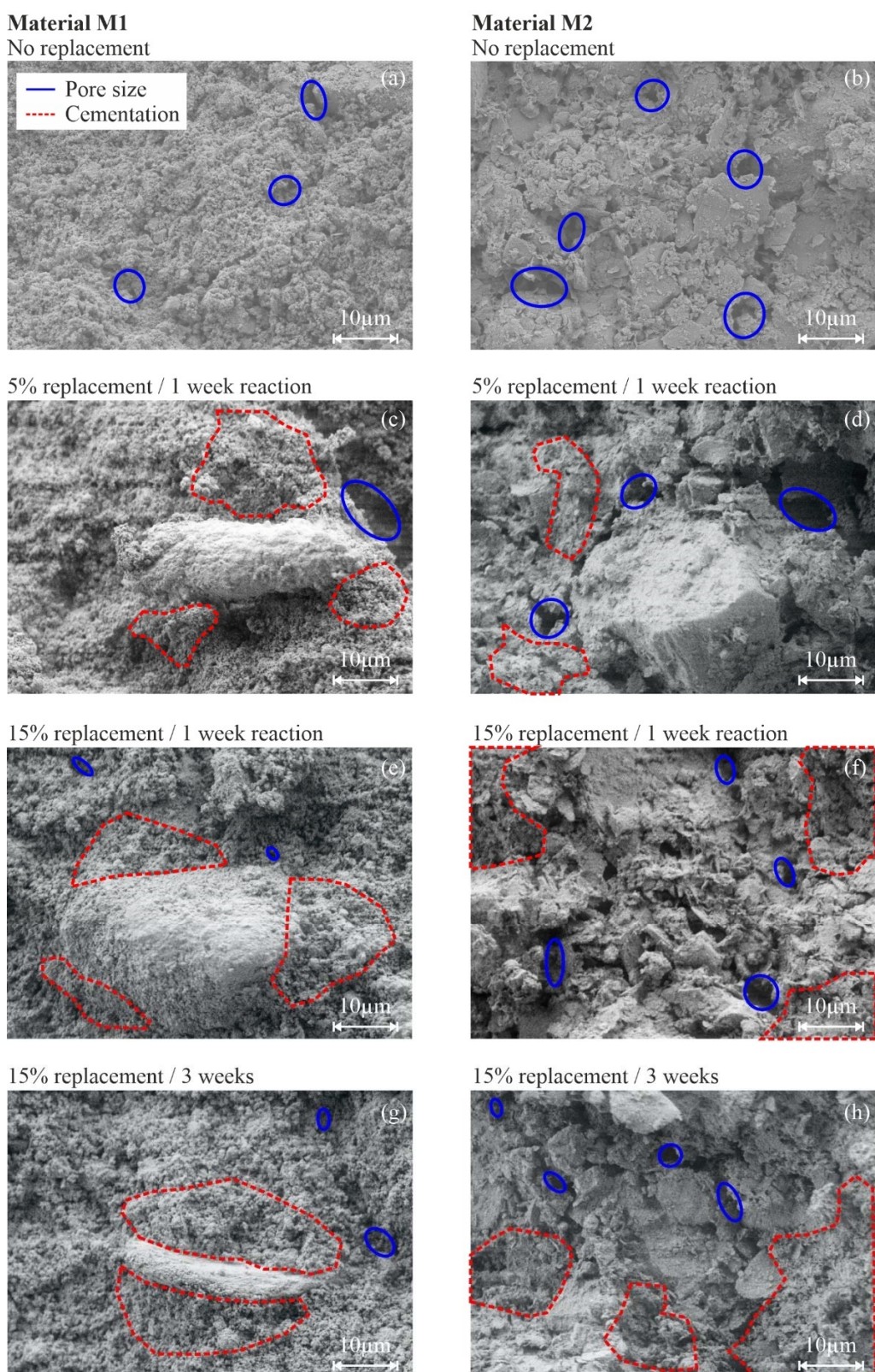

**Figure 8.** Scanning electron microscopy images of uncemented and artificially cemented soil specimens (*M1:* (**a,c,e,g**) column and *M2:* (**b,d,f,h**) column).

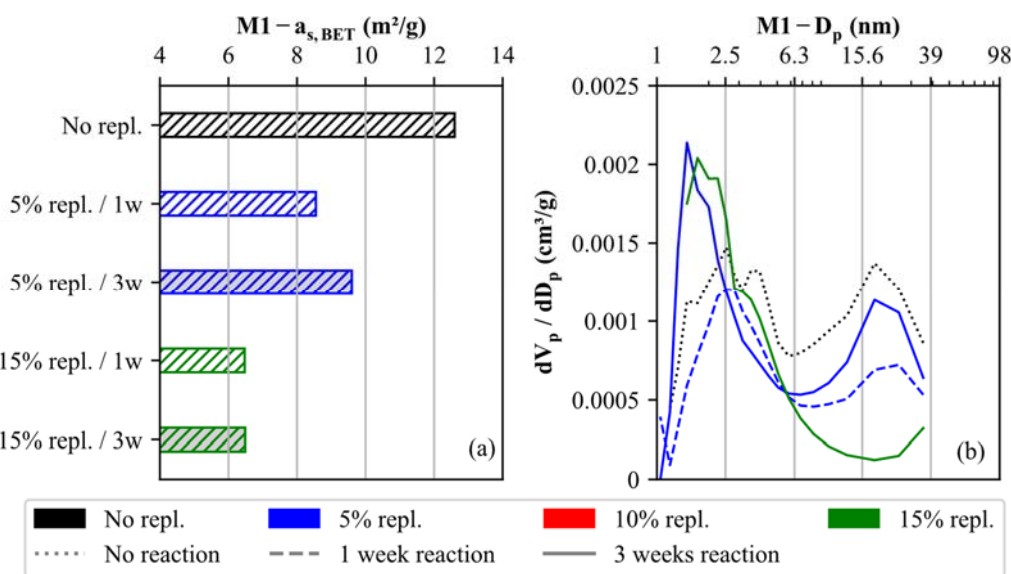

**Figure 9.** Comparison of (**a**) specific surface areas ($a_{s,BET}$) and (**b**) pore size distributions for uncemented and artificially cemented soil specimens (*M1*) determined by $N_2$-*BET* physisorption.

In addition, the pore size distribution of soil specimens is compared for *M1* in Figure 9b. The differential pore area (*dVp/dDp*) varies between 0.0007 and 0.0015 cm$^3$/g for pore sizes ranging from 1.3 to 35 nm. The local maxima are reached at pore sizes equal to 2.5 and 18 nm. This pore size distribution changes as soon as nano-calcite cements are formed. While the number of pore sizes >6 nm decreases, the number of small pores <6 nm significantly increases, corroborating the (nano/micro) filler effect. This trend is most evident in the soil specimen with 15% replacement cured for 3 weeks (green, solid line in Figure 9b).

The presented procedure of $CaCO_3$ formation has a significantly lower $CO_2$ footprint compared to lime and cement-based binders, considering the highly alkaline reaction, which leads to the absorption of $CO_2$. In contrast, larger replacements and higher material costs are required to reach the desired properties. For the practical application of soil stabilization work, it can be concluded that mineralogy has a strong influence on the success of the respective binder. Therefore, geotechnical laboratory and in-situ testing should be extended by mineralogical investigations at an early project stage.

## 6. Conclusions

Oedometer and bender element results, assisted by mineralogical (*XRD*) and microstructural (*SEM* and *BET-SSA*) data, proved that the formation of nanocrystalline calcite ($CaCO_3$) cements within the grain-to-grain matrix of either carbonate- or silicate-dominated artificial soils leads to an increase in the apparent preconsolidation pressure and shear wave velocity. However, this increase is strongly influenced by the mineralogical composition of soil specimens, the degree of $CaCO_3$ cementation and the curing time. If the bond strength is exceeded, a decrease in constrained modulus (*M*) and shear wave velocity ($V_S$) is obtained due to the destructuration of the soil's fabric. As interparticle bonding is intact, constrained moduli of cemented specimens (15% replacement) increased by factors ranging between 5 and 7 compared to the uncemented specimens. The increase in shear wave velocity ($V_S$) is less pronounced and approximates 1.1 for both lithologies, even though the silicate-rich soil sample is characterized by a softer behavior. It was observed that the mean effective stress (*p'*) has a stronger influence on the shear wave velocity ($V_S$) compared to deviatoric stresses (*q*). With respect to soil stabilization by means of artificial $CaCO_3$ cementation, it was shown that comparatively high amounts of chemical additives are needed to influence the stiffness of soft, fine-grained soils remarkably. However, the presented technique offers a more favorable $CO_2$ balance compared to lime and cement. The observations made are valid for carbonate- and silicate-rich soils; future studies are required to prove these trends

for different, more complex, mineralogical compositions and particle size distributions. Furthermore, the presented procedure of soil stabilization should be applied at different test sites to validate its possibilities and limitations for practical engineering.

**Author Contributions:** Conceptualization, S.O., A.B., M.D. and R.M.; methodology, S.O., R.M. and A.B.; software, S.O., A.B., D.M.M.T. and M.N.; validation, S.O. and A.B.; formal analysis, S.O. and A.B.; investigation, S.O., A.B., D.M.M.T. and M.N.; resources, R.M. and M.D.; data curation, S.O. and A.B.; writing—original draft preparation, S.O. and A.B.; writing—review and editing, S.O., A.B., F.T. and R.M.; visualization, S.O.; supervision, S.O. and A.B.; project administration, S.O.; funding acquisition, S.O. and R.M. All authors have read and agreed to the published version of the manuscript.

**Funding:** This research was funded by the Austrian Research Promotion Agency (FFG), grant number FO999891282.

**Institutional Review Board Statement:** Not applicable.

**Informed Consent Statement:** Not applicable.

**Data Availability Statement:** The data presented in this study are available on request from the corresponding author.

**Acknowledgments:** We thank Stefanie Eichinger, Andrea Wolf, Christine Latal, Andreas Hasawend, Daniel Vidonja and Anton Kaufmann for their invaluable support and help during laboratory testing. Open Access Funding by the Graz University of Technology.

**Conflicts of Interest:** The authors declare no conflict of interest.

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
