# Peer review of "Microstructure Development in Artificially Cemented, Fine-Grained Soils"

_geosciences, doi:10.3390/geosciences12090333_

Round 1
Reviewer 1 Report
The paper studies the microstructure of artificially cemented soils. In no case is the type of soil indicated in the FAO classification or similar, nor is the reason for its selection justified.
The paper needs to be rewritten as materials are mixed with methods and methods are not carefully described.
Also:
Lines 90-98: present a map indicating the locations cited.
Lines 96-100: The mineralogical analyses, how have they been obtained? If they are current, they should go to results.
Lines 101-118: The results have not been indicated how they have been achieved and the nature of the two samples studied has not been described, nor why they have been chosen.
Author Response
Point 1: The paper studies the microstructure of artificially cemented soils. In no case is the type of soil indicated in the FAO classification or similar, nor is the reason for its selection justified. The paper needs to be rewritten as materials are mixed with methods and methods are not carefully described.
Response 1: We thank the reviewer for the criticism. In the revised manuscript, we have provided the rationale for choosing the two soil mixtures, and added a definition for the term soil (i.e., defined here as fine-grained, postglacial sediments). Materials and Methods have been separated and are either discussed in more detail or explicitly referenced.
Point 2: Also: Lines 90-98: present a map indicating the locations cited.
Response 2: We don`t find it necessary to show these local locations in a separate map, as they are only exemplarily used to describe the mineralogical composition of typical postglacial (fine-grained) sediments from Austrian Alpine regions (Oberhollenzer, 2022). In our study, we have used artificial samples for testing, as described in the following sentences.
Oberhollenzer, S. (2022). Charakterisierung postglazialer, feinkorndominierter Sedimente mittels in-situ und Laborversuchen. PhD Thesis, TU Graz, Institute of Soil Mechanics, Foundation Engineering and Computational Geotechnics (supervisor: Prof. Marte).
Point 3: Lines 96-100: The mineralogical analyses, how have they been obtained? If they are current, they should go to results.
Response 3: The materials M1 and M2 represent typical soil mineralogies found in many Alpine regions, which we explicitly state in the text. The composition of the materials was verified by prior mineralogical analysis (XRD). This has been clarified in the revised version of the paper.
Point 4: Lines 101-118: The results have not been indicated how they have been achieved and the nature of the two samples studied has not been described, nor why they have been chosen.
Response 4: We partly agree with the comment. We have now linked the text with the methodology section for clarity. However, regarding the second comments, we want to add, that we explicitly state that our chosen test materials resemble natural materials found in Alpine postglacial settings.
Reviewer 2 Report
The authors needed to improve their manuscript in order to be processed further. Below are some of my comments:
· Abstract:
· The authors should provide the reason for performing their study or/and the importance of their study.
· Add some numerical results
· Introduction:
· Lines 46-49: Add some literature of reviews related to this section such as:
o Influence of soil stabilizing materials on lead polluted soils using Jet Erosion Tests. International Journal of Integrated Engineering, 9(1), 28-38.
o Predicting Mechanistic Detachment Model due to Lead-Contaminated Soil Treated with Iraqi Stabilizers. KSCE Journal of Civil Engineering, 23(7): pp 2898–2907. DOI: 10.1007/s12205-019-2312-3.
· Lines 62-64: Add some worldwide of literature of reviews related to the role of CaCo3 in soil stabilization such as:
o Behavior of soil erodibility parameters due to biological soil crusts using jet erosion tests. Ecological Engineering, 153: 105903. DOI: 10.1016/j.ecoleng.2020.105903.
o Quantifying mechanistic detachment parameters due to humic acids in biological soil crusts. Land, Volume 10, Issue 11, Article number 1180.
· The authors should also present the importance of this work. What is new in their work? What is the new science in their manuscript?
· Materials and methods:
· Lines 101-118: This section should be re-written and the phrase “Error! Reference source not found” should be replaced by appropriate words.
· Put the soil classification in Table 1.
· Make a flowchart of how to do your work.
· Results and Discussion:
· Lines 261-377: Again, these sections should be re-written and the phrase “Error! Reference source not found” should be replaced by appropriate words.
· Conclusions: The limitations of the study should be presented in this part. Also, future work should be presented in this part.
Author Response
Point 1: The authors needed to improve their manuscript in order to be processed further. Below are some of my comments.
Response 1: We thank the reviewer for the overall positive evaluation of our study.
Point 2: Abstract: The authors should provide the reason for performing their study or/and the importance of their study.
Response 2: The formation vs damaging of microstructure in postglcial, fine-grained sediments are poorly understood yet. We state this now in the Abstract. Add some numerical results. Done.
Point 3: Introduction: Lines 46-49: Add some literature of reviews related to this section such as: Influence of soil stabilizing materials on lead polluted soils using Jet Erosion Tests. International Journal of Integrated Engineering, 9(1), 28-38.; Predicting Mechanistic Detachment Model due to Lead-Contaminated Soil Treated with Iraqi Stabilizers. KSCE Journal of Civil Engineering, 23(7): pp 2898–2907. DOI: 10.1007/s12205-019-2312-3.
Response 3: Done.
Point 4: Lines 62-64: Add some worldwide of literature of reviews related to the role of CaCo3 in soil stabilization such as: Behavior of soil erodibility parameters due to biological soil crusts using jet erosion tests. Ecological Engineering, 153: 105903. DOI: 10.1016/j.ecoleng.2020.105903.; Quantifying mechanistic detachment parameters due to humic acids in biological soil crusts. Land, Volume 10, Issue 11, Article number 1180.
Response 4: Done.
Point 5: The authors should also present the importance of this work. What is new in their work? What is the new science in their manuscript?
Response 5: We have provided the rationale and objectives of our work in the second last paragraph of the Introduction.
Point 6: Materials and methods: Lines 101-118: This section should be re-written and the phrase “Error! Reference source not found” should be replaced by appropriate words.
Response 6: We now refer explicitly to the methods we used whenever appropriate. The referencing has been improved accordingly.
Point 7: Put the soil classification in Table 1.
Response 7: Done.
Point 8: Make a flowchart of how to do your work.
Response 8: We present a flowchart summarizing our work at the end of the Methods.
Point 9: Results and Discussion: Lines 261-377: Again, these sections should be re-written and the phrase “Error! Reference source not found” should be replaced by appropriate words.
Response 9: The referencing has been improved accordingly.
Point 10: Conclusions: The limitations of the study should be presented in this part. Also, future work should be presented in this part.
Response 10: We have added the limitations and future research directions in the conclusions.
Reviewer 3 Report
The findings of this scientific research are very interesting for the field of soil stabilization.
Author Response
Point 1: The findings of this scientific research are very interesting for the field of soil stabilization.
Response 1: We thank the reviewer for the very positive feedback we received.
Round 2
Reviewer 1 Report
The methods and materials sections remain unclear. They have lightened slightly.
In line 110 it says “as determined by prior mineralogical analysis”. Indicate where.
Author Response
Point 1: The methods and materials sections remain unclear.They have lightened slightly. In line 110 it says “as determined by prior mineralogical analysis”. Indicate where.
Response 1: Thank you very much for your review. We state this now in the Materials section (see Lines 108 - 109).
Reviewer 2 Report
The authors addressed well all my comments
Author Response
Point 1: The authors addressed well all my comments.
Response 1: Thank you very much for your review.